# RMS Delay Spread vs. Coherence Bandwidth from 5G Indoor Radio Channel Measurements at 3.5 GHz Band

**DOI:** 10.3390/s20030750

**Published:** 2020-01-29

**Authors:** Wout Debaenst, Arne Feys, Iñigo Cuiñas, Manuel García Sánchez, Jo Verhaevert

**Affiliations:** 1Department of Signal Theory and Communications-atlanTTic Research Center, University of Vigo, 36310 Vigo, Spain; woutdebaenst@live.be (W.D.); arne.feys@outlook.com (A.F.); inhigo@uvigo.es (I.C.); manuel.garciasanchez@uvigo.es (M.G.S.); 2IDLab, Department of Information Technology, Ghent University-imec, 9052 Gent, Belgium

**Keywords:** indoor propagation, modelling, radio propagation, ray-tracing

## Abstract

Our society has become fully submersed in fourth generation (4G) technologies, setting constant connectivity as the norm. Together with self-driving cars, augmented reality, and upcoming technologies, the new generation of Internet of Things (IoT) devices is pushing the development of fifth generation (5G) communication systems. In 5G architecture, increased capacity, improved data rate, and decreased latency are the objectives. In this paper, a measurement campaign is proposed; we focused on studying the propagation properties of microwaves at a center frequency of 3.5 GHz, commonly used in 5G cellular networks. Wideband measurement data were gathered at various indoor environments with different dimensions and characteristics. A ray-tracing analysis showed that the power spectrum is dominated by the line of sight component together with reflections on two sidewalls, indicating the practical applicability of our results. Two wideband parameters, root mean square delay spread and coherence bandwidth, were estimated for the considered scenarios, and we found that they are highly dependent on the physical dimension of the environment rather than on furniture present in the room. The relationship between both parameters was also investigated to provide support to network planners when obtaining the bandwidth from the delay spread, easily computed by a ray-tracing tool.

## 1. Introduction

Wireless communication systems have become one of the most interesting areas in the field of telecommunications and networking. With its exponential growth, they support networks, having a large impact on the daily lives of billions of people. In the last few decades, wireless communication has revolutionized the way in which people work and communicate, and even how they form social relationships with each other.

The speed of wireless data transfers is always being pushed beyond its limits. One of the upcoming technologies is fifth generation (5G), providing a solution for major practical communication problems for the Internet of Things (IoT) and augmented reality [1].

The performance of a digital communication system depends largely on the quality of the transmission channel it uses [2,3]. One of the main factors that limits the capacity of a certain system is inter-symbol interference [4,5]. A good knowledge of the different band parameters regarding frequency selectivity and the time dispersion is necessary to avoid this problem. With this aim, this paper contains the results of a measurement campaign investigating different wideband characteristics in the 3.5 GHz band. The interest in this band is growing, as indicated by the worldwide spectrum auctions performed in the spring and summer of 2018 for allowing the occupancy of this spectrum slice by 5G systems [6,7].

Many papers have already analyzed these channel characteristics for outdoor, outdoor to indoor, and indoor environments at different frequencies [8,9,10,11,12,13]. The results presented in these previous works were compared to those obtained in our measurement campaign to connect the contribution of this work with the previously reported research. Particularly, few have studied channel characteristics for indoor settings at the 3.5 GHz band [14,15,16,17]. In most of these cases, the number and characteristics of the environments and/or the measurement campaign designs differ considerably from these presented in the next sections. Other papers considering the same frequency band focused on parameters or characteristics that are not considered in the present contribution [18]. Thus, the provided insight could be useful for complementing the scientific knowledge in the field, and for extending and generalizing the conclusions. Completing this previous research, this paper describes a measurement campaign and discusses different insights, including useful conclusions on the relationship between the most important wideband parameters: root mean square (RMS) delay spread in the time domain and coherence bandwidth in the frequency domain.

The measurements focused on the influence of environmental changes on the channel response, considering effects such as the size of the room or the presence of furniture and other objects in standard offices. The setup consisted of a vector network analyzer (VNA), two omnidirectional antennas, and an automated rail positioner that moves the receiver antenna to different positions within the room. An automated measurement system, developed with MATLAB® (MathWorks, Natick, MA, USA), operated both the VNA and the rail positioning system. This system allowed the analysis of wideband information and the connection of these data to the specific environmental situations.

In addition, a simple ray-tracing tool provided insight into the different propagation mechanisms observed in measurement outcomes. This helped to explain the origin of most of the multipath components gathered at the reception point, and allows the extension of conclusions to other environments different from those strictly measured. The results of this paper will assist with the design of wireless communication systems operating around the 3.5 GHz band, as we provide the measured parameters and, even more importantly, strategies for interpreting the different environments. The large amount of gathered data allows the analysis of the relationship between RMS delay spread and coherence bandwidth in a variety of environments and conditions, representing insight into future developments.

This paper follows this organization: Section 2 presents the automated measurement system together with information about the measurement environments, as well as the theoretical fundamentals applied to obtain the wideband parameters and to extend the applicability range. The results of the measurement campaign are provided in Section 3. Section 4 analyzes the measured outcomes, providing insight into the measured outcomes, and indicates how to extend the conclusions to other environments using ray-tracing analysis. Finally, Section 5 presents the conclusions.

## 2. Materials and Methods

### 2.1. Measurement Setup

An Agilent Fieldfox N9913A vector network analyzer (VNA) was used to execute the measurements to obtain the frequency response of the radio channel, providing complex S_21_ parameter values. This scattering parameter S_21_ is a measure of the amplitude attenuation and the phase difference of the measured channel between two ports (transmitter and receiver) of the VNA, including cabling, antennas, and the propagation channel. Knowing that the band around 3.5 GHz is the most popular for 5G communications among those below 6 GHz in the current trials, tests, and initial experiences [19], the VNA gathered the measurements in that part of the spectrum, more specifically in the 3.35–3.65 GHz frequency band.

The Fieldfox was connected to two azimuth-omnidirectional biconical wideband antennas (Electro-Metrics EM-6865, Johnstown, NY, USA) via two appropriate coaxial cables with N-connectors. The transmitting (Tx) antenna was attached to the ceiling, while the receiving (Rx) one was placed on a pole at a height of 1.1 m. Both antennas were vertically polarized. This setup imitates cellular phone indoor picocells, with base station antennas at higher heights and mobile terminals carried by their user around a height of 1.1 m. To analyze the effect of the user position, the receiving antenna was moved along a positioning rail driven by a stepper motor. Thus, the antenna could be moved with a precision of less than 1 mm, impossible to be performed manually. The measurement setup is depicted in Figure 1.

Both the movement of the receiver along the positioner and the frequency sweep of the VNA were controlled by a laptop. This device had a transmission control protocol/Internet protocol (TCP/IP) connection over an unshielded twisted pair connection to the VNA and a serial connection over a recommended standard 232 (RS232)-computer serial interface (IEEE) line to the stepper motor on the positioning rail. The measurement process was fully automated to reduce the risk of human mistakes, to record the obtained data in a structured manner, and to reduce measurement time. The program operating the complete flow of the measurement ran on a Unix-based system using the programming environment MATLAB® (MathWorks, Natick, MA, USA). Via Standard Commands for Programmable Instruments (SCPI) commands over the TCP/IP connection, the laptop set the VNA to the following initial settings:Mode = Network Analyzer (NA)Center frequency = 3.5 GHzFrequency bandwidth = 300 MHzOutput power = 3 dBmReceiver IF bandwidth = 10 kHzAveraging = 10Samples/Sweep = 1001

Once everything was initialized, the receiving antenna was moved to the start position and then the measurement began. After taking 10 frequency sweeps and averaging the measurement results, the antenna was moved over one-quarter of a wavelength at center frequency, more specifically, over 0.021 m, and then a new measurement was performed at this position. The measured channel responses were saved in Touchstone (*.s2p) format locally on the VNA and later copied to MATLAB® (MathWorks, Natick, MA, USA) for further processing. This process was repeated until the receiving antenna reached the end position.

To measure distances larger than the range of the positioner, the rail was moved further, with the receiving antenna starting on the spot where it previously stopped. Hence, the measurements recorded by the positioner from different locations could be concatenated, resulting in measurements over a longer distance.

The averaging performed by the VNA over 10 samples for each receiving position was the first step to reduce the noise and the temporary effects in the environment, contributing to lower measurement uncertainty. Another possible measurement error source comes from the VNA, for which standard uncertainty of the S_21_ parameter is below 0.3 dB in amplitude and 2 degrees in phase.

### 2.2. Measurement Environment

The measurement campaign was performed at the School of Telecommunication Engineering (University of Vigo, Vigo, Spain), involving four distinct rooms: a corridor, a large auditorium, an electronics laboratory, and an office. Taking into account the different configurations (line of sight, furnished and unfurnished, and the receiver path orientations), up to 10 different environments are investigated. Table 1 summarizes the environment dimensions and the measurements performed in each of them, under line of sight (LoS), obstructed line of sight (OLoS), and non-line of sight (NLoS).

Figure 2 depicts the corridor, where two measurements took place. In the first one, the transmitter was attached to the ceiling (2.50 m high), indicated by position Tx1, and the receiver started underneath it, moving away from transmitter, following the central axis of the corridor, as also indicated by an arrow in the floorplan. A constant uninterrupted LoS path between the receiver and transmitter was guaranteed. Next, the transmitter was placed at position Tx2 and the receiver was moved along the same path. This allowed the LoS between the transmitter and receiver to be interrupted at a distance of 2 m on the rail, and thus measurements were under NLoS conditions from that point. This corridor is located under the ground level, so its walls are made of reinforced concrete. This is especially important when analyzing the NLoS scenario, as the blocking of LoS is considerable.

Figure 3 contains a map of the auditorium. The measurements were executed in two directions, indicated by arrows on the floorplan. Firstly, the receiver on the rail started underneath the transmitter, which was attached to the ceiling (2.90 m high) at position Tx on the floorplan, moving away from this location. Secondly, the receiver moved in an opposite direction compared to the previous experiment. 

Figure 4 depicts the floorplan of the lab. Many objects were present in the room, which formed an interesting and realistic setting for indoor wideband measurements. There were desks and wardrobes composed of wood, iron machinery, and a variety of chairs, computers, and other office equipment. The LoS conditions were not guaranteed strictly, as in many paths the Fresnel ellipsoids around the direct link between transmitter and receiver were disrupted by the different elements within the room. Thus, OLoS better described the environment in both configurations, labeled ID1 and ID2. Two receiving antenna rail setups, orthogonal to each other, were used at the receiver side during the measurements, whereas the transmitter was located at a fixed position of 30 cm below the ceiling.

The floorplan of the office environment can be found in Figure 5. This office helped us to analyze the impact of furniture regarding channel characteristics. Thus, measurements were recorded with furniture as depicted in Figure 5, and then the same room was unfurnished to obtain frequency responses within an empty room. Both situations were LoS, as the direct path between transmitter and receiver was unobstructed for the duration of the experiment: the presence of furniture (mainly office elements: desks, chairs, and computer equipment) simply added more possible propagation paths, as new obstacles surrounded the antennas. At the receiving end, two antenna rail setups were used, also orthogonal to each other. Each measurement setup was executed twice: in furnished and unfurnished situations. 

Importantly, during the execution of these measurements, no people were allowed to cross the different environments to minimize time variability as much as possible. Thus, the assumption of quasi-stationary conditions during the measurements is realistic.

### 2.3. Theoretical Fundamentals

The S_21_ parameters, measured between the two ports of the VNA, not only include the measured channel but also the effects of the cables and antennas. To obtain the S_21_ parameters solely of the radio channel (only the antennas and the propagation channel), the system was calibrated via a through-connection. Using Equation (1), the frequency response of the system *H*_system_(*f*) was subtracted from the measured complex frequency response *H*_measured_(*f*) to obtain the channel frequency response *H*_channel_(*f*) exclusively [4,20,21,22].
(1)$$H_{channel}\left(f\right)=\frac{H_{measured}\left(f\right)}{H_{system}\left(f\right)}$$

The inverse Fourier transform provides the impulse response of the channel in the delay domain.
(2)$$h\left(\tau \right)=\int_{-\infty }^{+\infty }H_{channel}\left(f\right)\cdot e^{j2\pi f\tau }df$$

Note that measurements were only recorded over a limited bandwidth (covered by the frequency sweep of the measurements); thus, a windowing function was applied with *f*_1_ and *f*_2_ equal to 3.35 and 3.65 GHz, respectively.
(3)$$W\left(f\right)=\{\begin{array}{c} H_{system}\left(f\right)\;\;\;\;\;\;\;\;\mathrm{if}\;f_{1}\leq f\leq f_{2}\;\\ 0\;\;\;\;\;\;\;\;\;\;\;\;\;\;\mathrm{otherwise} \end{array}$$

The measured result is hereby an estimation of the frequency response of the channel:(4)$$H_{measured}\left(f\right)=H_{channel}\left(f\right)\cdot W\left(f\right)$$
(5)$$h_{measured}\left(\tau \right)=\int_{-\infty }^{+\infty }H_{channel}\left(f\right)\cdot W\left(f\right)\cdot e^{-j2\pi f\tau }df.$$

Noise was already reduced by the VNA by averaging over 10 samples, but some noise was still present in the measurement outcomes. To discern the different multipath reflections from the remaining noise, a side lobe level (SLL) technique was applied to classify all values underneath this threshold as noise and set them to zero. As the used window was rectangular, an SSL value of 13 dB lower than the maximum value (generally related to the direct path) was applied [4].

The parameters commonly used to describe the wideband behavior of a channel are the RMS delay spread (τ_RMS_) and the coherence bandwidth (Bc). Both are based on the power delay profile, which, in quasi-static environments, can be obtained by taking the average of the squared impulse response of Equation (5). This averaging was executed in this work by a sliding window over 11 samples in which the value in the middle was replaced by the result [23]:(6)$$P_{h}\left(\tau \right)=\langle {\left|h\left(\tau \right)\right|}^{2}\rangle$$

With the power delay profile, wideband parameters, such as mean delay spread (τ_mean_) and τ_RMS_, can be calculated. These are popular factors that portray the time dispersion of a channel. The τ_RMS_ is, in other words, an estimation of the minimum time between transmission of symbols, necessary to avoid inter-symbol interference caused by multipath propagations [24,25,26].
(7)$$\tau_{mean}=\frac{\sum_{k}P_{h}\left(\tau_{k}\right)\tau_{k}}{\sum_{k}P_{h}\left(\tau_{k}\right)}$$
(8)$$\tau_{rms}=\sqrt{{\bar{\tau }}^{2}-{\left(\tau_{mean}\right)}^{2}}$$
where:(9)$${\bar{\tau }}^{2}=\frac{\sum_{k}P_{h}\left(\tau_{k}\right)\tau_{k}^{2}}{\sum_{k}P_{h}\left(\tau_{k}\right)}.$$

Next to the effects time dispersion has on a channel, the frequency selectivity must be considered. A commonly used parameter is *B_c_*, which is an estimation of the maximum bandwidth that is available without fading. This value can be obtained by first calculating the frequency correlation function by taking the inverse Fourier transform of the power delay profile.
(10)$$R_{T}\left(\Delta f\right)=\int_{-\infty }^{+\infty }P_{h}\left(\tau \right)\cdot e^{-j2\pi \Delta f\tau }d\tau$$

The coherence bandwidth is the approximation of the maximum bandwidth over which two frequencies components exhibit a correlated behavior [26,27,28]. Commonly used correlation levels are 0.5, 0.7 and 0.9 for which the corresponding coherence bandwidth are B_0.5_, B_0.7_, and B_0.9_.
(11)$$B_{c}=min\left(\Delta f\right)\;\;\;\mathrm{such}\;\mathrm{that}\;\;\left|R_{T}\left(\Delta f\right)\right|=c$$

In the literature, the relationship between coherence bandwidth and RMS delay spread was investigated. The proposed relationship follows an inverse equation [3]:(12)$$B_{c}=\frac{1}{\alpha \cdot \tau_{rms}}$$
where α can be estimated by experimental analysis. Then, different values were proposed, showing enough differences to support investigating this parameter in the 3.5 GHz band in different indoor environments. A value of α = 2π was the result of a theoretical analysis of exponentially decaying power delay profiles [24]. For indoor channels, α can vary between small values and 10 [10]. In OLoS situations, α tends to be smaller than in LoS situations [29]. Using a two-ray model of the impulse response, a value α = 6 was estimated [30]. More recently, a different relationship driven by two parameters was proposed, given an additional degree of freedom for the analysis [16]. Besides, this parameter can be related to the limit provided by [31], which can be written as α ≤ 2π/arccos (*c*), where *c* being the coherence level. 

### 2.4. Ray-Tracing Simulations

Ray-tracing techniques [32,33,34] were used to compute the path of electromagnetic waves in order to know more about the influence of external objects such as walls, ceilings, and floors, and their effect on the channel. To obtain more insight in the channel response, a straightforward ray-tracing tool was programmed for determining the most important contributions to the multipath patterns in the different environments. The tool calculated the delay of some main rays using the positions of both Rx and Tx antennas, considering the dimensions of the room.

The values for the different parameters characterizing the building materials present in the various environments can easily be obtained from scientific literature. Depending on the frequency band, the variability of local materials, and the theoretical model used to extract the electromagnetic parameters from complex channel response measurements, the data provided present large differences [35,36,37,38,39,40,41,42,43,44,45]. Regardless, the interest in ray-tracing analysis along this work lies more in the insight this technique provides on the multipath components generated by the room geometry than the exact amplitude of each multipath component. This means that ray-tracing allows us to identify the origin of each measured contribution in the delay domain and, comparing the predicted component with the measured data for the same time delay, to determine the actual influence of different room elements. In such circumstances, the exact amplitude of each component was not the objective of this work, and the precise characterization of the building materials was beyond the focus in this study.

After inspection of the measurement environments and the involved antennas, we considered that there would be a minor effect of waves reflecting on the floor or ceiling and then reaching the receiving antenna. The antenna was omnidirectional in the azimuth and had a 3 dB beam width of 50° in elevation. This resulted in neglecting the reflections on the ceiling and floor for the considered rail positions.

## 3. Results

After the measurement campaign, the outcomes were complex frequency responses of the radio channel, with 1001 frequency spots at each receiving point, separated 0.021 m from each other, with this volume of data in each of the 10 different environments. This volume of data does not provide clear insight into the behavior of the 3.5 GHz indoor radio channel. Thus, the application of the mathematical processing introduced in Section 2 was mandatory for interpreting and analyzing the experimental results.

The first action was the computation of power delay profiles, which helped in understanding the different contributions to the multipath signals of the environmental elements. In this computation, the windowing was considered, as indicated in the previous section, using the SLL suppression technique [3,23].

Figure 6, Figure 7 and Figure 8 plot the results in the auditorium and corridor, representing the power delay profiles in color codes. Each color map represents the power received, normalized by its maximum within each environment, as a function of the time delay in the abscissa axis (i.e., the power delay profile), at each receiver location indicated as a distance along the linear positioner in the ordinate axis. The dominant contributions provided by the ray-tracing tool are depicted as black lines to provide clarity in identifying the origin of each multipath element.

In Figure 6, the power delay profile at 500 cm from the beginning of the receiver path is represented, as this is the traditional method of showing such results. The color maps allow the presentation of a collection of power delay profiles in only one figure, so this was the selected method of showing the results.

Although approximate models exist to predict or generate power delay profiles [46], they explain reasonable LoS environments. However, OLoS and NLoS situations are not well modeled, as the random disposition of furniture and operative elements within a room is not easily predicted. Then, measured data represent a must when analyzing such scenarios.

Figure 6 and Figure 7 represent the auditorium results with the receiver moving away or transversally, respectively. The ray-tracing results helped with interpreting the data given by the plot. Accordingly, the shortest delay contribution in Figure 6 corresponds to line of sight; the second one to a single reflection on the black board installed in front of the room; the third, on the side walls; the fourth relates to double reflections on side walls; and the last one corresponds to the reflection on the wall opposite to the black board.

As expected, the results are more symmetric when measuring along the transverse direction. In this case (Figure 7), the first contribution arriving in delay was again due to the line of sight; the second from the black board; the third and fourth, in a cross shape, correspond to each of the side walls in a symmetric effect due to the symmetry of the configuration; and the fifth is due to the opposite wall.

Figure 8 represents the LoS situation of the channel in the corridor. The first contribution, in this case, is related to the line of sight; the second is a double reflection on the side walls; in the middle, there are components coming from the side walls; and the third is related to the opposite wall (in this case, completely consisting of a metallic door).

Data on the delay domain allowed the computation of τ_mean_ and τ_RMS_, which are the most interesting parameters for planning wideband communication systems, as their values are meaningful for controlling and avoiding inter-symbol interference. To provide a confident value, Table 2 contains the measured RMS delay spreads, for which 90% of the cases were lower than the given value, and for each of the environments. Thus, it is guaranteed that RMS delay spreads only surpassed the given value in 10% of the measured locations.

We observed that transverse paths provided shorter RMS delay spreads than those paths with the receiver moving away from the transmitter antenna. This occurred in all considered environments except in the empty office, as even that environment is almost squared and the difference in delay is minimal.

Coherence bandwidth values at a correlation level 0.9 are summarized in Table 3. Values varied from 5.8 MHz in the corridor (LoS conditions) to 13.4 MHz in the laboratory.

## 4. Discussion

Several kinds of results are provided in this paper: those directly from measurements (as the coherence bandwidth and RMS delay spread values), those from ray-tracing simulations, and the relation between wideband parameters, which represents the main contribution of this work.

Coherence bandwidth and RMS delay spread values provide direct indications to network planners when deploying a 5G network in indoor environments. Coherence bandwidth was around 10 to 13 MHz in most of the considered environments, excepting for a long and narrow corridor, where it decayed to 5.8 MHz, and also a symmetric and large auditorium, with values between 6.6 to 8.0 MHz. The presence of furniture does not seem to induce important variations, as differences were less than 2% of the coherence bandwidth. Comparing these results to those previously published, they demonstrate consistency [16], as previous research gives around 16 MHz with coherence level 0.9 within the office area, and around 4 MHz at the auditorium.

All environments presented RMS delay spreads around 11 to 13 ns, except for the auditorium and the corridor under LoS conditions. In both cases, for short distances, the influence of the antenna pattern was noticeable. In the first 2.20 m away from the transmitter, in the auditorium and in the corridor under LoS conditions, the receiving antenna started underneath the transmitting antenna. Because the radiation pattern of both antennas was almost zero in this direct link, there were almost no line of sight rays from one to the other. This made the received power mostly due to reflections and thus resulted in a higher RMS delay spread, which explains the larger values of these environments. The effect of blocking the line of sight in the corridor was also noticeable, as commonly RMS delay spreads are higher in NLoS than in LoS, and here the results were lower. The strong attenuation provided by the blocking walls (a reinforced concrete corner in the basements, rounded by the terrain) ensured that all contributions arriving at the receiver did so after at least two reflections on different walls, and, many times, after three or more reflections. The attenuation induced by those reflection events and the absence of a direct path with short delay forced the concentration of contributions with low amplitude, and then a small RMS delay spread. A similar situation was also reported in [16], where the authors explained that multipath richness occurs in environments with symmetric configurations and lack of furniture, which was the case in our corridor.

The obtained RMS delay spreads in the offices are coherent with those provided by [15], who reported values from 8 to 10 ns after an ultra-wideband measurement campaign, covering the spectrum from 3.1 to 10.6 GHz, which includes the frequency under study. The indoor environment, in that case, was a 63 m^2^ furnished office room. Measurements within a smaller auditorium reported RMS delay spreads between 17.9 and 34.5 ns, depending on the distance to the transmitter, in a band from 3 to 4 GHz [16], with our auditorium results being in good agreement. Larger RMS delay spreads, up to 45 ns, were reported in [17] for a complex office environment. The variety of environments presented in the current paper extends the validity of those more limited contributions, providing information regarding a wider collection of possible scenarios and including comparisons on the effect of furniture or soft walls.

At lower frequencies, larger RMS delay spreads have been reported. At 2.35 GHz, delays of 21.6 ns in LoS conditions and between 30 and 43 ns in NLoS were provided [12]. We also observed RMS delay spreads of 26 ns in corridors and 19 and 17 ns in a room (LoS and NLoS, respectively) at the 5 GHz band [13], which are in similar magnitude of orders as those obtained in our research, but at higher frequencies. At 11 GHz, values of 20 ns were reported in NLoS rooms [11]. Even larger values for indoor-to-outdoor scenarios were recorded [10], between 50 and 100 ns, probably influenced by the attenuated contribution of the direct propagation path compared to the effect of the other multipath components. In that research, at 2.45 and 5.2 GHz, no significant differences were detected between both bands [10]. These frequencies are on both sides of the frequency considered in the present paper, which supports the coherence of our data with those of other previous studies.

The result of the ray-tracing analysis provides insight in the effect of each constructive element of the considered environments, as the identified contributions of each multipath component from different boundary walls and elements explain the different peaks of the average power delay profiles computed from the measured data. Knowing more about the emergence of the measurement results allows analyses that are more thorough. The approximation of the indoor measurements with a simple ray-tracing tool opens the possibility of using such a limited model to predict and design new deployments in the band around 3.5 GHz. As simulations explain the measured results, ray-tracing appears to be a good tool for simulating such kinds of environments. A simple tool (just two reflections in azimuth and one reflection in elevation), combined with precise building material characterization, seems to be enough to provide valid predictions of the measured environment and thus to be used in planning new indoor deployments.

The value of α in Equation (12) can be used in estimating the coherence bandwidth from the RMS delay spread, which can be computed using simple ray-tracing. An example of the application of that equation is depicted in Figure 9. Hence, an exact relationship between coherence bandwidth and RMS delay spread does not exist [47]. An inverse relation has been reported [48]. Thus, there is a high degree of uncertainty regarding the estimation of the coherence bandwidth based on the RMS delay spread, and this paper provides an estimation of this relationship in the 3.5 GHz band in different indoor environments, as shown in Table 4.

All obtained values for α were within the range from 2.2 to 6.4, and most of them were around 6. This corresponds to the literature: α = 2π (approximately 6.28) was proposed in [29], α = 6 in [30], and α < 10 in [21]. This consistence with parameters obtained at different frequencies reinforces the interest in such a relationship. α tends to be smaller in OLoS conditions than in LoS. Values at 3.5 GHz, provided in Table 4, provide information for 5G radio network planners when they deal with different indoor environments.

## 5. Conclusions

This paper presents results related to a large wideband measurement campaign, performed in 10 indoor environments and configurations, at a frequency related to 5G systems: the band from 3.35 to 3.65 GHz, resulting in a 300 MHz bandwidth around 3.5 GHz. The interest in this band, and the upcoming explosion in 5G application for indoor domestic and professional users, reinforces the applicability of the insights provided throughout this manuscript and summarized in these conclusions.

We focused on the wideband parameter values that guarantee the performance of at least 90% of the cases. In such circumstances, RMS delay spreads between 11 to 13 ns seem to be reasonably reachable in most of the considered environments, which varied in size, shape, and furniture occupancy. The coherence bandwidth at a correlation level 0.9 was between 10 and 13 MHz. With both parameters, an initial assessment of future 5G environments was made. Further, we observed that furniture does not add large variations to these wideband parameters, which simplifies the planning of 5G indoor networks.

The complex wideband measured data also allowed us to test a simple ray-tracing tool. With the help of this simulation tool, the origin of most of the multipath contribution was explained, and the main conclusion is that simple ray-tracing is enough for explaining most situations. Considering the direct path and one or two reflections on the walls was enough in most of the environments. Using azimuth omnidirectional antennas, the ceiling and floor do not provide important contributions. So, with just one reflection on any of these boundaries, the simulation would be enough for an accurate approximation.

Ray-tracers directly provide data related to delay spread. The relation between RMS delay spread and coherence bandwidth was analyzed, providing values related to both parameters. Thus, coherence bandwidth can easily be estimated from a calculated RMS delay spread, completing the wideband prediction in synthetic environments.

As furniture does not induce significant variations in wideband parameters; a ray-tracing simulation in empty environments would provide a good approximation of the real world in a 5G indoor radio channel at the 3.5 GHz band. This is critical information for network designers, as this simplifies the planning of indoor 5G networks.

## Figures and Tables

**Figure 1 sensors-20-00750-f001:**
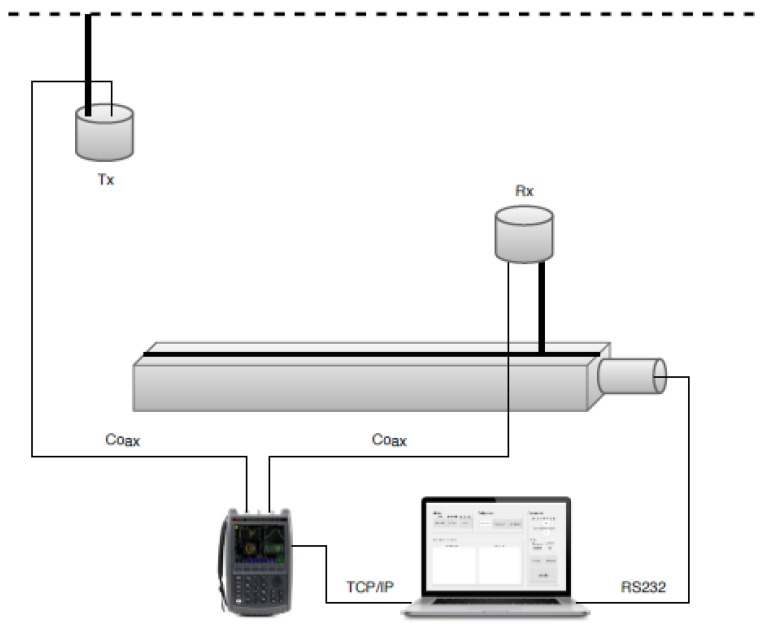
Scheme of the measurement setup.

**Figure 2 sensors-20-00750-f002:**
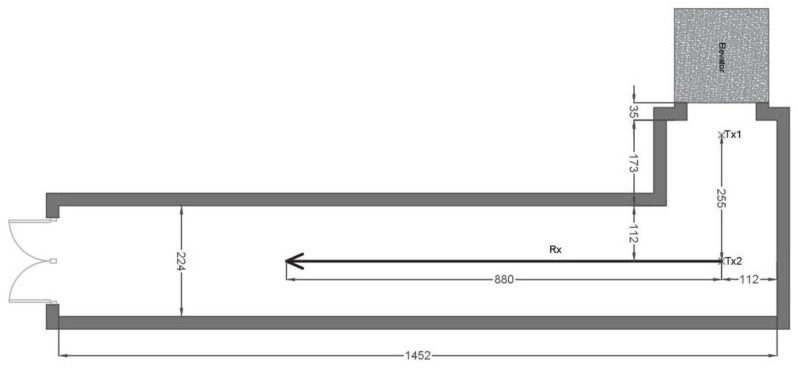
Floorplan of the corridor with dimensions in cm.

**Figure 3 sensors-20-00750-f003:**
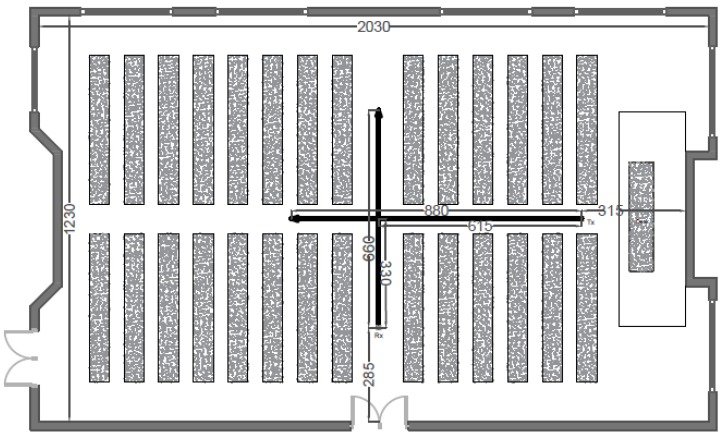
Floorplan of the auditorium with dimensions in cm.

**Figure 4 sensors-20-00750-f004:**
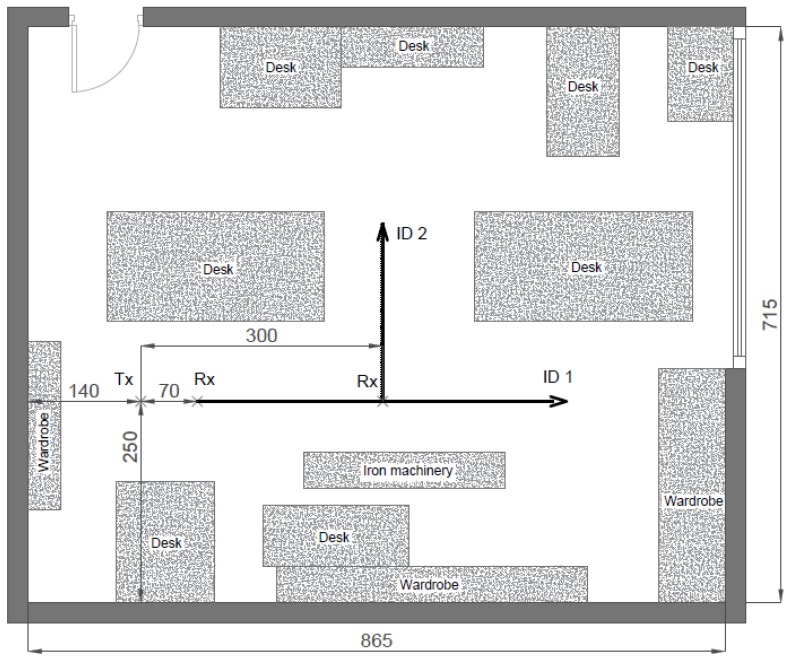
Floorplan of the laboratory with dimensions in cm.

**Figure 5 sensors-20-00750-f005:**
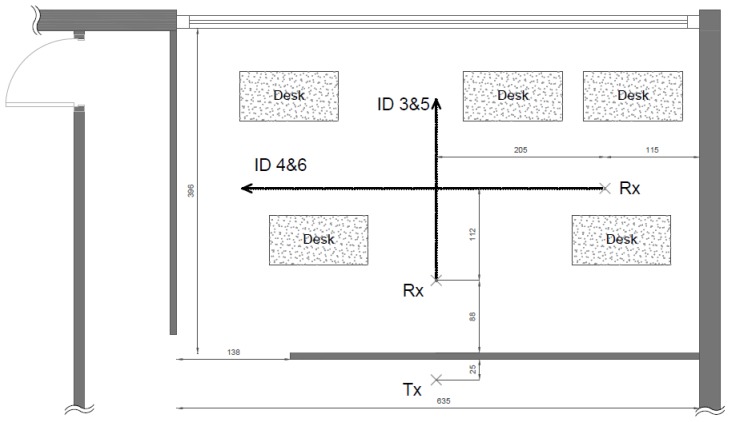
Floorplan of the office (furnished) with dimensions in cm.

**Figure 6 sensors-20-00750-f006:**
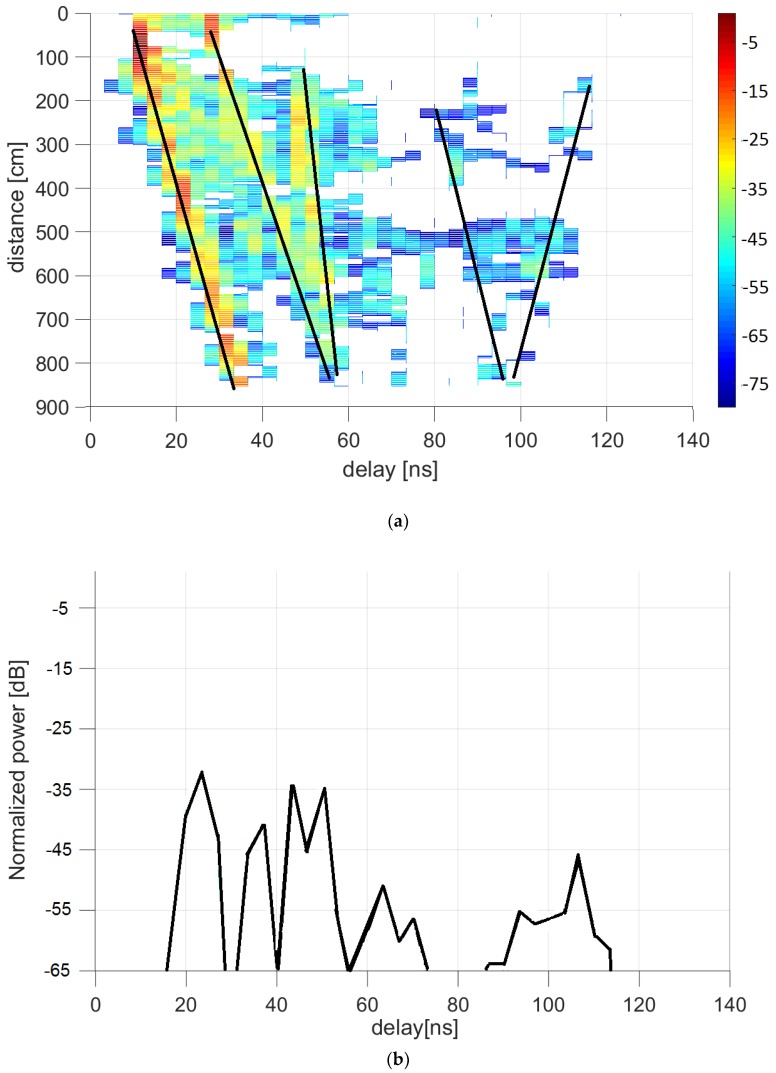
Power delay profile at auditorium with receiver moving away from transmitter: (**a**) measurements (in dB relative to maximum) and ray-tracing analysis; (**b**) representation at 500 cm from the beginning of the rail.

**Figure 7 sensors-20-00750-f007:**
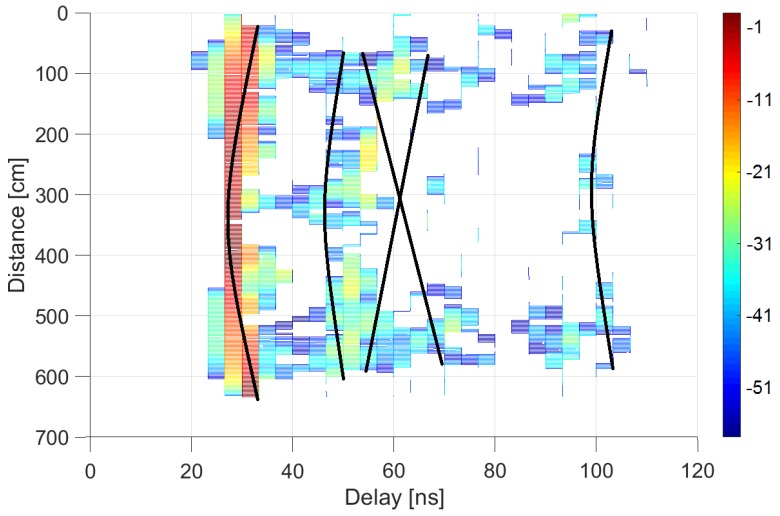
Power delay profile at auditorium with receiver moving in transverse direction: measurements (in dB relative to maximum) and ray-tracing analysis.

**Figure 8 sensors-20-00750-f008:**
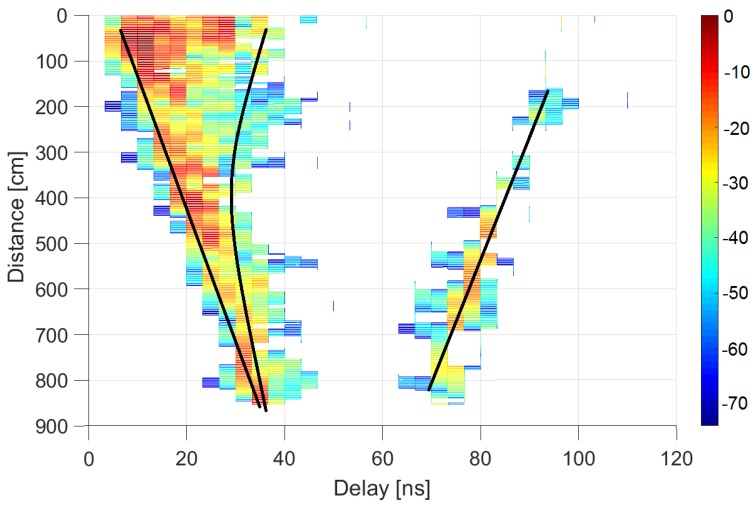
Power delay profile in a corridor under LoS conditions: measurements (in dB relative to maximum) and ray-tracing analysis.

**Figure 9 sensors-20-00750-f009:**
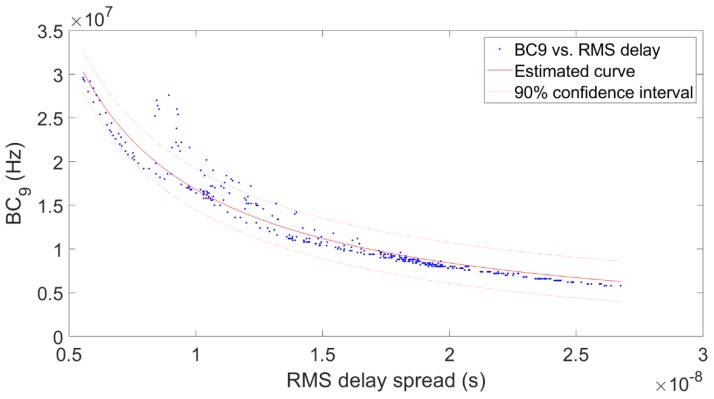
Coherence bandwidth at correlation level 0.9 vs. RMS delay spread in the auditorium with the receiver moving away from the transmitter.

**Table 1 sensors-20-00750-t001:** Summary of measurement environments in different line of sight conditions: line of sight (LoS), obstructed line of sight (OLoS) and non-line of sight (NLoS).

Environment	Room Size (m)		Path Length (m)	Line of Sight Conditions
	Width	Length		
Corridor	2.24	14.52	8.80	LoS
				NLoS
Auditorium	12.30	20.30	8.80	LoS (away)
			6.69	LoS (transverse)
Laboratory	7.15	8.65	4.40	OLoS (away)
			2.20	OLoS (transverse)
Office (furnished)	3.95	6.35	2.20	LoS (away)
			4.40	LoS (transverse)
Office (unfurnished)			2.20	LoS (away)
			4.40	LoS (transverse)

**Table 2 sensors-20-00750-t002:** Measured root mean square (RMS) delay spread (ns), for which 90% of cases are lower.

Environment	Line of Sight Conditions	RMS Delay Spread (ns)
Corridor	LoS	26.2
	NLoS	15.4
Auditorium	LoS (away)	23.6
	LoS (transverse)	20.4
Laboratory	OLoS (away)	19.4
	OLoS (transverse)	13.4
Office (furnished)	LoS (away)	14.6
	LoS (transverse)	11.6
Office (unfurnished)	LoS (away)	12.3
	LoS (transverse)	11.7

**Table 3 sensors-20-00750-t003:** Measured coherence bandwidth (MHz) at correlation level 0.9, for which 90% of cases are larger.

Environment	Line of Sight Conditions	Coherence Bandwidth (MHz)
Corridor	LoS	5.8
	NLoS	11.4
Auditorium	LoS (away)	6.6
	LoS (transverse)	8.0
Laboratory	OLoS (away)	10.6
	OLoS (transverse)	13.4
Office (furnished)	LoS (away)	13.2
	LoS (transverse)	12.6
Office (unfurnished)	LoS (away)	13.0
	LoS (transverse)	12.6

**Table 4 sensors-20-00750-t004:** Factor relating coherence bandwidth at correlation level 0.9 and RMS delay spread.

Environment	Line of Sight Conditions	α
Corridor	LoS	4.6
	NLoS	5.4
Auditorium	LoS (away)	5.9
	LoS (transverse)	2.2
Laboratory	OLoS (away)	4.9
	OLoS (transverse)	5.6
Office (furnished)	LoS (away)	6.3
	LoS (transverse)	6.2
Office (unfurnished)	LoS (away)	6.4
	LoS (transverse)	6.2

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
