# Peer review of "RMS Delay Spread vs. Coherence Bandwidth from 5G Indoor Radio Channel Measurements at 3.5 GHz Band"

_sensors, 2020, doi:10.3390/s20030750_

Round 1
Reviewer 1 Report
The paper deals with fundaments of the today’s wireless communication, which the coherence bandwidth relation to the RMS delay spread surely is. The conclusions are supported with measurements, which is always beneficial. However many papers on this topic has been published already and many of them are not referenced. The relation to existing work is therefore a problematic aspect of this manuscript, while a credible justification of another paper on this topic is just missing. I am not going to provide you with list of papers which are dealing with the same topic and showing virtually the same figures, because this could be seen as a misconduct, but I think that proper literature review needs to be done and the obtained results needs to be thoroughly related to the state of the art and their importance shall be evident.
Author Response
Reply to Reviewer 1
Thank you for your valuable remarks. These are without doubt very helpful to improve the quality of our paper. In the following, the reply to your comments (in italic) can be found as well as the corresponding changes that we have made to the paper.
Remark 1: The paper deals with fundaments of the today’s wireless communication, which the coherence bandwidth relation to the RMS delay spread surely is. The conclusions are supported with measurements, which is always beneficial.
Thank you for this positive feedback.
Remark 2: However many papers on this topic has been published already and many of them are not referenced. The relation to existing work is therefore a problematic aspect of this manuscript, while a credible justification of another paper on this topic is just missing. I am not going to provide you with list of papers which are dealing with the same topic and showing virtually the same figures, because this could be seen as a misconduct, but I think that proper literature review needs to be done and the obtained results needs to be thoroughly related to the state of the art and their importance shall be evident.
Thank you for this valuable comment. We related the results from our measurement campaign to results of existing work, described in the literature. Therefore two references are added to the reference list: [12] and [13] (new numbering) and the following text has been added to the paper: “The results presented in these previous works are compared to those obtained in our measurement campaign, in order to connect the contribution of this work with previously reported research.” And in the reference list:
[12] Zhang, N.; Wang, H.; Hong, W.; Zhou, J.; Yang, G.; Zang, H.; Yu, C. Investigations on wideband MIMO indoor channel characteristics at 2.35GHz with multiple polarized antennas. In the Proceedings of the 2012 IEEE International Symposium on Antennas and Propagation, 2012.
[13] Zhao, X.; Li, S.; Liang, X.; Wang, Q.; Hentila, L.; Meinila, J. Measurements and modelling for D2D indoor wideband MIMO radio channels at 5 GHz. IET Communications. 2016, vol. 10, no. 14, pp. 1839-1845.
This literature study also resulted in the following publications [15], [16], [17] and [18] (new numbering), which have been added to the reference list. However, the number and characteristics of the environments and/or the design of the measurement campaign differed considerably with our approach. Reference [18] uses the same frequency bands, but focuses on other parameters and characteristics. We added therefore the text in red from line 51 to line 56: “In most of these cases, the number and characteristics of the environments and/or the measurement campaign designs differ considerably from these along the next sections. Other papers, in the same frequency band, focus on parameters or characteristics that are not considered in the present contribution [18]. Thus, the provided insight could be useful in complementing the scientific knowledge in the field, and to extend and generalize the conclusions.”
The reference list has been complemented with the following papers:
[15] Haneda, K.; Richter, A.; Molisch, A.F. Modeling the frequency dependence of ultra-wideband spatio-temporal indoor radio channels. IEEE Trans. Ant. Propag. 2012, vol. 60, no. 6, pp. 2940-2950.
[16] Pérez, J.R.; Torres, R.P.; Rubio, L.; Basterrechea, J.; Domingo, M.; Rodrigo Peñarrocha, V.M.; Reig, J. Empirical characterization of the indoor radio channel for array antenna systems in the 3 to 4 GHz frequency band. IEEE Access. 2019, vol. 7, pp. 94725-94736.
[17] Huang, 489 F.; Tian, L.; Zheng, Y.; Zhang, J. Propagation characteristics of indoor radio channel from 3.5 GHz to 28 GHz. In the Proc. IEEE 84th Veh. Technol. Conf., Montreal, 2016, Quebec City, Canada.
[18] Torres, R.P.; Pérez, J.R.; Basterrechea, J.; Domingo, M.; Valle, L.; González, J. ; Rubio, L.; Rodrigo, V.M.; Reig, J. Empirical characterisation of the indoor multi-user MIMO channel in the 3.5 GHz band. IET Mw, Ant. & Propag. 2019, vol.13, iss.13, pp.2386-2390.
Besides the relationship between the coherence bandwidth and RMS delay spread from [3], another relationship is referenced [31] (new numbering), giving an additional degree of freedom for the analysis. Additional text has been written: “More recently, a different relationship driven by two parameters has been proposed, given an additional degree of freedom for the analysis [16]. Besides, this parameter can be related to the limit provided by [31], which can be written as α ≤ 2π /arccos(c), being c the coherence level.”
Reference [31] (new numbering) has been added to the reference list: [31] Fleury, B. H. An uncertainty relation for WSS processes and its application to WSSUS systems. IEEE Trans. Commun. 1996, vol. 44, no. 12, pp. 1632–1634.
As already mentioned above, due to this extensive literature study, our results are mapped with the obtained results in the references. Therefore, in Section 4. Discussion, we added in red: “Comparing these results to previously published ones, they show a good agreement with [16], that gives around 16 MHz with coherence level 0.9 within the office area, and around 4 MHz at the auditorium.” and “The obtained RMS delay spreads at offices are coherent with those provided by [15], that gives values from 8 to 10 ns after an ultra-wideband measurement campaign, covering the spectrum from 3.1 to 10.6 GHz, which includes the frequency under study. The indoor environment, in that case, was a 63 square meter furnished office room. Measurements within a smaller auditorium reported RMS delay spreads between 17.9 and 34.5 ns, depending on the distance to the transmitter, in a band from 3 to 4 GHz [16], being our auditorium results in good agreement. Larger RMS delay spreads, up to 45 ns, are reported in [17] for a very complex office environment. The variety of environments presented in the current paper extends the validity of those more limited contributions, providing information regarding a wider collection of possible scenarios, and including comparisons on the effect of furniture or soft walls.
At lower frequencies, larger RMS delay spreads have been reported. At 2.35 GHz, delays of 21.6 ns in LoS conditions and between 30 and 43 ns in NLoS are provided at [12]. We can also observe RMS delay spreads of 26 ns in corridors and 19 and 17 ns in a room (LoS and NLoS, respectively), at the 5 GHz band [13], which are in similar magnitude orders of those obtained in our research, but at higher frequencies. At 11 GHz, values of 20 ns are reported in NLoS rooms [11]. Even larger values for indoor-to-outdoor scenarios are given [10], between 50 and 100 ns, probably influenced by the attenuated contribution of the direct propagation path compared to the effect of the other multipath components. What is interesting in this research, done at 2.45 and 5.2 GHz, is that no significant differences have been detected between both bands [10]. These are at both sides of the frequency considered in the present paper, which supports the idea of the coherence of our data with other previous studies.’
Reviewer 2 Report
The authors have studied the delay spread and coherence bandwidth based on indoor channel measurements. It is interesting that they have obtained the measurements not only from one scenario but several scenarios by regular moving of the receiver. The results can be interesting for the communication community. The paper lack of new theoretical results.
In general, the paper is well written, but some minor changes must be done, for examples:
All abbreviations must be defined, for examples, IoT (in the Abstract) and RMS.
Page 1: “…different band parameters” seems not to be a scientific way of writing
Page 2:”..using the understanding ..” seems not to be a scientific way of writing
The “complex S_21 parameter” must be defined.
Author Response
Reply to Reviewer 2
Thank you for your valuable remarks. These are without doubt very helpful to improve the quality of this paper. In the following, the reply to all your comments (in italic) can be found as well as the corresponding changes that we have made to the paper.
Remark 1: The authors have studied the delay spread and coherence bandwidth based on indoor channel measurements. It is interesting that they have obtained the measurements not only from one scenario but several scenarios by regular moving of the receiver. The results can be interesting for the communication community.
Thank you for this positive feedback.
Remark 2: The paper lack of new theoretical results.
The focus of this paper is an extensive measurement campaign, which studies the propagation properties of a commonly used 5G cellular network frequency band. The obtained wideband measurement data at various indoor environments is analyzed by the root mean square (RMS) delay spread and the coherence bandwidth. The highly dependency on the environment (and not on the furniture present in the room) and the relationship between RMS delay spread and coherence bandwidth are some key results of this paper. With our measurement campaign, we hence support the existing theory, while limiting the calculations in practice. Delay spreads, calculated by a simple ray-tracing tool (where furniture can be left out), result very efficiently in bandwidths, which can be used by 5G network planners.
Remark 3: In general, the paper is well written, but some minor changes must be done, for examples:
All abbreviations must be defined, for examples, IoT (in the Abstract) and RMS.
Thank you for your comment. We checked the complete manuscript and added the full text every first time a new abbreviation is mentioned. Hence, we added in the abstract and the rest of the text fourth generation (4G), Internet of Things (IoT), fifth generation (5G) and root mean square (RMS), as is also indicated in red in the manuscript itself.
Remark 4: Page 1: “…different band parameters” seems not to be a scientific way of writing
Thanks for the suggestion. We removed the word band, because the relation to the frequency band is elaborated more in the sentences thereafter. It results in the following sentence (also in red in the manuscript itself): “A good knowledge of the different parameters regarding frequency selectivity and the time dispersion is necessary to avoid this phenomenon.”
Remark 5: Page 2:”..using the understanding ..” seems not to be a scientific way of writing
Thank you for your remark. We replaced the words “using the understandings in” by the word “interpreting”. It results in the sentence, which is colored red in the manuscript: “The results of this paper will help the design of wireless communication systems operating around the 3.5 GHz band, as it gives measured parameters and, even more important, strategies for interpreting different environments.”
Remark 6: The “complex S_21 parameter” must be defined.
Thank you for your comment. We added the following text (shown in red) on line 79 and further to the manuscript: “This scattering parameter S21 is a measure for the amplitude attenuation and the phase difference of the measured channel between the two ports of the VNA (including cabling, antennas and propagation channel itself).”
2.11.0.0Round 2
Reviewer 1 Report
My major comments are tackled.